# Long-Read Whole Genome Sequencing Elucidates the Mechanisms of Amikacin Resistance in Multidrug-Resistant *Klebsiella pneumoniae* Isolates Obtained from COVID-19 Patients

**DOI:** 10.3390/antibiotics11101364

**Published:** 2022-10-06

**Authors:** Andrey Shelenkov, Lyudmila Petrova, Anna Mironova, Mikhail Zamyatin, Vasiliy Akimkin, Yulia Mikhaylova

**Affiliations:** 1Central Research Institute of Epidemiology, Novogireevskaya Street 3a, Moscow 111123, Russia; 2National Medical and Surgical Center Named after N.I. Pirogov, Nizhnyaya Pervomayskaya Street 70, Moscow 105203, Russia

**Keywords:** antimicrobial resistance, *Klebsiella pneumoniae*, COVID-19, plasmids, multidrug resistance, genomic epidemiology, whole genome sequencing

## Abstract

*Klebsiella pneumoniae* is a Gram-negative, encapsulated, non-motile bacterium, which represents a global challenge to public health as one of the major causes of healthcare-associated infections worldwide. In the recent decade, the World Health Organization (WHO) noticed a critically increasing rate of carbapenem-resistant *K. pneumoniae* occurrence in hospitals. The situation with extended-spectrum beta-lactamase (ESBL) producing bacteria further worsened during the COVID-19 pandemic, due to an increasing number of patients in intensive care units (ICU) and extensive, while often inappropriate, use of antibiotics including carbapenems. In order to elucidate the ways and mechanisms of antibiotic resistance spreading within the *K. pneumoniae* population, whole genome sequencing (WGS) seems to be a promising approach, and long-read sequencing is especially useful for the investigation of mobile genetic elements carrying antibiotic resistance genes, such as plasmids. We have performed short- and long read sequencing of three carbapenem-resistant *K. pneumoniae* isolates obtained from COVID-19 patients in a dedicated ICU of a multipurpose medical center, which belonged to the same clone according to cgMLST analysis, in order to understand the differences in their resistance profiles. We have revealed the presence of a small plasmid carrying *aph(3′)-VIa* gene providing resistance to amikacin in one of these isolates, which corresponded perfectly to its phenotypic resistance profile. We believe that the results obtained will facilitate further elucidating of antibiotic resistance mechanisms for this important pathogen, and highlight the need for continuous genomic epidemiology surveillance of clinical *K. pneumoniae* isolates.

## 1. Introduction

The antimicrobial resistance (AMR) of pathogenic and opportunistic bacteria, especially in clinical settings, has become a major challenge that threatens the success of different protection measures in various medical applications [1]. Currently, drug-resistant infections account for about 700,000 deaths globally, and this number may increase up to several millions in the next decades [2]. According to the pathogens priority list of the World Health Organization (WHO), carbapenem-resistant Enterobacteriaceae were assigned a critical level due to increasing morbidity and mortality caused by them [3]. Within this bacterial family, *Klebsiella pneumoniae* possessing resistance to carbapenems is the most common and dangerous species associated with mortality rates exceeding 30% [4,5]. *K. pneumoniae* is one of the leading causes of healthcare-associated infections worldwide, including sepsis, pulmonary diseases, and urinary tract infections [6].

The global influence of the COVID-19 pandemic on bacterial AMR is yet to be estimated, but some reports have already confirmed the increasing number of infections caused by bacteria producing extended-spectrum beta-lactamases (ESBLs) and carbapenemases [7,8]. The optimal treatment regimen for infections caused by carbapenem-resistant *K. pneumoniae* is yet to be developed [9,10], and the existing options include the administration of colistin, ceftazidime/avibactam, or meropenem/vaborbactam, in high doses [10,11], which are also prone to resistance development and are burdened with substantial toxicity profile [10].

In order to address these challenges appropriately, novel prevention strategies and treatment plans are required, and their development would hardly be possible without the investigation of AMR mechanisms and routes of resistance spreading within the bacterial population. The diffusion of resistance genes is usually attributed to horizontal gene transfer mediated by plasmids, and conjugated plasmids are recognized as important vectors for AMR gene transmission in Gram-negative bacteria [12,13]. In recent years, whole genome sequencing (WGS), especially long-read sequencing, has become a powerful tool for the determination of plasmid structures and AMR gene locations [14,15,16]. WGS also allows for performing a reliable epidemiological surveillance for outbreak investigations, in which it is vitally important to determine whether particular bacterial isolates belong to an outbreak-causing strain, or not [17].

In this work, we have performed short- and long read sequencing of three carbapenem-resistant *K. pneumoniae* isolates obtained from COVID-19 patients in a dedicated intensive care unit (ICU). Although these isolates constituted the same strain according to cgMLST analysis, their phenotypic AMR profiles were different. Hybrid short- and long-read assembly allowed us to reveal a small plasmid carrying *aph(3′)-VIa* gene providing resistance to amikacin in one of these isolates, which explained the difference in resistance. We believe that the results obtained will contribute to the understanding of antibiotic resistance mechanisms, both in general, and for this important pathogen in particular. Further investigations in this field will ultimately lead to developing better prevention strategies in hospital settings.

## 2. Results

### 2.1. Isolate Typing, Resistance Profile, AMR Gene and Plasmid Content Determination

The typing results revealed the same profile ST395/KL39/O1/O2v1 for all three isolates. cgMLST analysis revealed that the genomic sequences of all isolates were very close (no allele differences between CriePir335 and 336 and six different alleles between either of these isolates and CriePir342). Thus, according to the criterion described previously (less than 18 different cgMLST alleles for *K. pneumoniae* [18]), these isolates were highly likely representing a single strain. Complete cgMLST profiles for the isolates studied are given in Appendix A.

All isolates were multidrug-resistant (MDR) and susceptible only to ceftazidime/avibactam and amikacin (except CriePir342). They carried multiple antibiotic resistance determinants, including ESBL-coding genes *blaCTX-M-15*, *blaOXA-48* and *blaTEM-1B* (see Figure 1). The only difference in genomic resistance was revealed for CriePir342, that carried a *aph(3′)-VIa* providing resistance to amikacin [19], which corresponded perfectly with phenotypic data. In general, we have not revealed any discrepancies between phenotypic and genomic resistance profiles. The susceptibility to ceftazidime/avibactam can be attributed to the synergistic effect of this drug.

The isolates CriePir335 and CriePir336 possessed the same five plasmids, while CriePir342 included an additional IncQ1 plasmid. This plasmid carried the *aph(3′)-VIa* gene mentioned above, which provides resistance to amikacin [19]. Plasmid data are provided in Table 1.

We investigated IncQ plasmid more thoroughly to get additional insights into the resistance mechanisms. We revealed a Tn5393 transposon homology, as well as two copies of IS91 insertion sequence, and IS91 family transposase between them, near the *aph(3′)-VIa* gene. The plasmid also included the genes encoding replication proteins RepA, RepB and RepC, and mobilization protein genes *mobA* and *mobC*. The plasmid structure is shown in Figure 2.

### 2.2. Virulence Factors

The three isolates studied included exactly the same set of virulence factors. The list and brief description of important gene clusters are presented in Table 2. The complete list of 87 virulence genes is provided in Appendix A. Most genes were located on chromosomes, except for *iucABCD*, *iutA* and *rmpA2*, which were located on IncH1B virulence plasmid.

The virulence factor content of CriePir isolates does not allow assigning them to the hypervirulent type since the set of corresponding genes was limited and most heavy metal resistance genes like *pbr* (lead resistance)*,pco* (copper)*,ter* (tellurite) and *sil* (silver) were missing. However, the isolates deserve additional attention to the presence of mucoid phenotype regulator *rmpA2* and their rapid spread within ICU.

### 2.3. Comparison with Reference Isolates of ST395 from Genbank Database

The isolates obtained were compared to the genomes of ST395 available in Genbank, based on cgMLST. A list of closest matches and their description are provided in Table 3, and the minimum spanning tree for these isolates is shown in Figure 3 All comparisons were made based on genome sequences as the phenotype data were not available.

AMR gene content was similar for reference and CriePir isolates, except for *aac(3)-IIa* found in the USA isolate only, *aph(3′)-VIa* revealed in CriePir342 and GCA_013421315 only, and *ant* genes (see Appendix A). An important difference was also that GCA_009661195 did not include *blaCTX-M-15* and *blaTEM-1B* beta-lactamase genes, although it possessed IncR plasmid, on which these genes were located in CriePir342.

Virulence factor sets were also quite similar for all isolates, but several noticeable differences were revealed. GCA_003401055 (USA), GCA_009661195 (Russia) and GCA_022988285 (Germany) lack the genes from the *iuc* cluster and *rmpA2*. These genes were located on IncH1B plasmid in CriePir342, and the first two reference isolates did not have this plasmid, while the isolate from Germany did. However, this is not surprising since the plasmid structure has a high degree of plasticity.

Only the isolate GCA_013421315 included IncQ1 plasmid, which was likely to carry *aph(3′)-VIa*. Unfortunately, the genome record did not contain the plasmid structure, so the direct comparison of plasmid sequences was impossible. However, the contig containing the *aph(3′)-VIa* AMR gene was mapped to the plasmid IncQ1 of CriePir342 with 100% identity (data not shown). At the same time, GCA_013421315 included four additional plasmids in comparison to CriePir342.

The isolates provided in the Table 3 constitute a single strain, or clone, with CriePir342 based on criteria proposed by Schursch et al. [18]. The closest genomic matches to our isolates were revealed in Germany, Finland and the USA. However, the plasmid content of these isolates was different from the one of CriePir342 (see Appendix A), namely, the isolates from Finland and Germany did not include IncQ1 plasmid, and the USA isolate possessed two additional plasmids, ColRNAI and IncL.

## 3. Discussion

In this study, a genomic epidemiology investigation of three MDR *K. pneumoniae* isolates from the patients of a dedicated COVID-19 ICU was conducted. Long-read sequencing allowed us to reveal that although these isolates constituted one strain, one of them (CriePir342) had a slightly different resistance profile and possessed additional plasmid encoding *aph(3′)-VIa* providing resistance to amikacin, which corresponded perfectly to its phenotype.

Currently *K. pneumoniae* is the most common cause of nosocomial infections in Russia, accounting for almost 30% of cases in 2020 according to the AMRmap database [20] (https://amrmap.ru/, accessed on 23 August 2022). The isolates studied belonged to the ST395 group, which is known as a high-risk clone with high capacity of drug resistance acquisition [21] and was revealed in different countries, for example, in France [22], Hungary [23] and China [24]. Additionally, ST395/KL39 isolates carrying the *blaOXA-48* gene were recently revealed in Russia, including CriePir234 (GCA_009661195.1) described earlier by us [25], as well as in Finland in the same year, and were also obtained in 2013–2016 in such distant regions of the world as China, Germany and the USA. All these isolates belonged to the same strain as our isolates according to the proposed cgMLST allele difference threshold (≤18, [18]), and contained similar sets of AMR and virulence genes. The AMR determinants included ESBL-encoding genes *blaOXA-48*, *blaCTX-M-15* (except GCA_009661195.1) and *blaTEM-1B* (except GCA_009661195.1), as well as various other AMR genes sufficient to consider these isolates as multidrug-resistant (MDR). However, only the Russian isolate NNKP343 (GCA_013421315.1) possessed the *aph(3′)-VIa* AMR gene mentioned above.

The virulence gene content was also similar for most isolates, except for the *rmpA2* encoding mucoid phenotype regulator, which was revealed in Russian isolates only, including the CriePir ones. Most virulence genes, except the *iuc* cluster and *rmpA2* mentioned above, were located on the chromosomes, which complies with previous data [25,26,27]. Another important determinant is a capsule surrounding the surface of *K. pneumoniae*, which serves as a main virulence factor associated with the viscous phenotype [28]. A capsular polysaccharide on the bacterial cell surface plays an important role in the pathogenicity of various bacteria, including *Acinetobacter baumannii* [29] and *K. pneumoniae* [30]. Although the KL39 capsule type is not generally considered as providing hypervirulence characteristics, at least one recent report described the increased virulence for the isolate having this capsule type and O1/O2v1 O-locus, which was found in all the isolates described above [31].

At the same time, the reference isolates exhibited differences in plasmid type and number, both between each other and with our set. For example, all isolates had IncR plasmid encoding, among others, the *blaOXA-1* beta-lactamase gene, but IncF1B was revealed in two Russian isolates only, while large IncHI1B pNDM-MAR-like plasmid carrying both resistance and virulence (*iuc* cluster) traits was revealed in all but two isolates. Last but not the least, IncQ1 plasmid, which carried the amikacin resistance gene in CriePir342, was revealed only in NNKP343 (GCA_013421315.1) reference isolate, thus allowing us to suppose, together with the fact that this isolate exhibited amikacin resistance [32], that this plasmid included *aph(3′)-VIa*. Unfortunately, the exact plasmid structures were not reconstructed in any reference isolates, and thus the direct comparison was not possible. However, additional analysis revealed that the IncQ1 plasmid mentioned above included fragments of the Tn5393 transposon that was known to carry aminoglycoside resistance genes (in our case, aminoglycoside-O-phosphotransferase), and partial sequences of which were more common than the complete transposon in plasmids and genomic islands [33]. We also revealed that this plasmid had a MOB_Q_ type and was mobilizable, rather than conjugative, which was reported to be a common characteristic of the IncQ1 plasmids [34]. Plasmids of this type are non-self-transmissible, but their host independent replication system allowed them to have a broader host range than any other known replicating components in bacteria [35]. Mobilizable plasmids rely on conjugative plasmids to provide the mating pair formation components, and MPF_T_ plasmids can serve as helpers in MOB_Q_ plasmid conjugation [36]. In CriePir342, conjugation could be assisted by large IncH plasmid, but more data are required to prove this hypothesis.

In general, the chromosomal DNA of *K. pneumoniae* carries only inherent resistance genes, and most acquired resistance determinants, including ESBLs, are located on plasmids [6,12,25]. Thus, it is not surprising that differences in plasmid carriage designated the dissimilarities in acquired AMR gene content for CriePir and reference isolates. It should be mentioned that although the reference and our isolates appeared to constitute a single clone, which supposedly spread across the world more than 10 years ago, the acquisition of virulence and AMR determinants through plasmid routes could significantly change the pathogenicity and morbidity of particular isolates. Meanwhile, the plasmid IncQ1 carrying the amikacin resistance gene was revealed in CriePir26 and CriePir28 isolates sequenced by us in 2017 [25]. Although these isolates belonged to a different clone with ST377, the plasmid sequence was completely the same as in CriePir342. Therefore, a continuous genomic epidemiology surveillance of clinical *K. pneumoniae* isolates is required to assess their threat to the patients of particular health care institutions since the determination of a sequence type and clonal lineage appears to be insufficient for this purpose.

The current study is limited to three isolates only since the main goal was to elucidate the difference in the phenotypic resistance profiles for the three *K. pneumoniae* isolates obtained in the same ward during this limited period. This was achieved with the help of third-generation sequencing. During recent years, the increasing use of long-read sequencing greatly facilitated the investigations of outbreaks and exploration of possible resistance transmission routes [37,38,39], and several bioinformatics protocols were developed for these purposes [15,17,40]. Another possible application of this powerful technology is the investigation of bacterial clone diversity within a particular hospital department or some other healthcare facility [41,42,43].

At the same time, various factors limit the ability to analyze the putative outbreak genomes in real-time, which, surprisingly, might include not the restrictions imposed by sequencing technologies, but rather data management and bioinformatics, for example, the lack of common outbreak repositories and delays between data collection and computational analysis [44,45]. Thus, creating more sophisticated bioinformatics tools can also advance AMR prevention strategies and the epidemiological surveillance of *K. pneumoniae* and other important pathogens. The development of such tools is one of the future perspectives of our investigations.

## 4. Materials and Methods

### 4.1. Sample Collection, Susceptibility Testing, DNA Isolation, and Sequencing

Three *K. pneumoniae* isolates (named CriePir335 (from urine), CriePir336 (from blood) and CriePir342 (from urine)) were obtained from COVID-19 patients in a dedicated ICU of a multipurpose medical center during the first half of June 2020. The patients were females of 81, 72 and 85 years of age, respectively, with a confirmed diagnosis of COVID-19.

Species identification for the isolates studied was performed using time-of-flight mass spectrometry (MALDI-TOF MS) with the VITEK MS (bioMerieux, Marcy-l’Étoile, France). Antimicrobial susceptibility/resistance was determined by the disc diffusion method using the Mueller-Hinton medium (bioMerieux, Marcy-l’Étoile, France) and disks with antibiotics (BioRad, Marnes-la-Coquette, France), and the minimum inhibitory concentration (MIC) was determined on VITEK 2 Compact 30 analyzer automated system (bioMerieux, Marcy-l’Étoile, France). The antibiotics tested included amikacin, amoxicillin/clavulanic acid, ampicillin, cefepime, ceftazidime, ceftriaxone, ceftazidime/avibactam, ciprofloxacin, ertapenem, fosfomycin, gentamicin, imipenem, levofloxacin, meropenem, netilmicin, tetracycline and trimethoprim/sulfamethoxazole. We used the EUCAST clinical breakpoints, version 11.0 (https://www.eucast.org/clinical_breakpoints/, accessed on 20 December 2020) to interpret the susceptibility/resistance results obtained.

The isolates described above represent a subset of multidrug-resistant (MDR) samples, which were subjected to WGS.

Genomic DNA was isolated using DNeasy Blood and Tissue kit (Qiagen, Hilden, Germany), and Nextera™ DNA Sample Prep Kit (Illumina^®^, San Diego, CA, USA) was applied for paired-end library preparation and WGS of the isolates on Illumina^®^ Hiseq 2500 platform (Illumina^®^, San Diego, CA, USA). The same genomic DNA was used to prepare the libraries for the Oxford Nanopore MinION sequencing system (Oxford Nanopore Technologies, Oxford, UK) with the Rapid Barcoding Sequencing kit SQK-RBK004 (Oxford Nanopore Technologies, Oxford, UK). The amount of initial DNA was 400 ng for each sample. The libraries were prepared according to the manufacturer’s protocols, and were sequenced on R9 SpotON flow cell with a standard 24 h sequencing protocol using the MinKNOW software (Oxford Nanopore Technologies, Oxford, UK).

### 4.2. Data Processing and Genome Assembly

The base calling of the raw MinION data was performed using the Guppy Basecalling Software version 4.2.2 (Oxford Nanopore Technologies, Oxford, UK), and demultiplexing was performed using the Guppy barcoding software version 4.2.2 (Oxford Nanopore Technologies, Oxford, UK). Hybrid short- and long-read assemblies were obtained using the Unicycler version 0.4.9 (normal mode) [46]. Genome assemblies were submitted to the NCBI Genbank under the project PRJNA839643.

The pipeline described earlier [25] was used for the assembled genome processing and annotation. The Resfinder 4.0 database was used for antimicrobial gene detection (https://cge.cbs.dtu.dk/services/ResFinder/, accessed on 20 August 2022, using default parameters). Virulence factors were revealed by searching in VFDB (http://www.mgc.ac.cn/VFs/main.htm, accessed on 20 August 2022, using default parameters). Plasmids were detected using the PlasmidFinder (https://cge.cbs.dtu.dk/services/PlasmidFinder/, accessed on 20 August 2022, using default parameters).

Isolate typing was performed by MLST using BIGSdb (https://bigsdb.pasteur.fr/klebsiella/, accessed on 25 May 2022). In addition, the types based on capsule synthesis loci (K-loci) and lipooligosaccharide outer core loci (OCL) were also deduced using the Kaptive software v.2.0.3 with default parameters [30].

The detection of cgMLST profiles was performed using the MentaList (https://github.com/WGS-TB/MentaLiST, version 0.2.4, default parameters, accessed on 25 August 2022) [47] using the scheme obtained from cgmlst.org (https://www.cgmlst.org/ncs/schema/schema/2187931/, contained 2,358 loci, last update 25 May 2022). The minimum spanning tree was built using PHYLOViz online (http://online.phyloviz.net, accessed on 20 August 2022).

We used the TnCentral database (https://tncentral.proteininformationresource.org/tn_blast.html, accessed on 20 August 2022) to search for transposon sequences and ISEscan [48] for searching insertion sequences.

## 5. Conclusions

We performed WGS and obtained hybrid short- and long-read assemblies of three MDR ESBL-producing *K. pneumoniae* isolates, representing a single clone, obtained from COVID-19 patients in a dedicated ICU. Bioinformatics analysis allowed us to elucidate the differences in their antibiotic resistance profiles and reveal the possible mechanism of amikacin resistance found in one of the isolates. We believe that our data will facilitate the understanding of transfer mechanisms and developing new strategies for preventing resistance spreading within the clinical *K. pneumoniae* population.

## Figures and Tables

**Figure 1 antibiotics-11-01364-f001:**
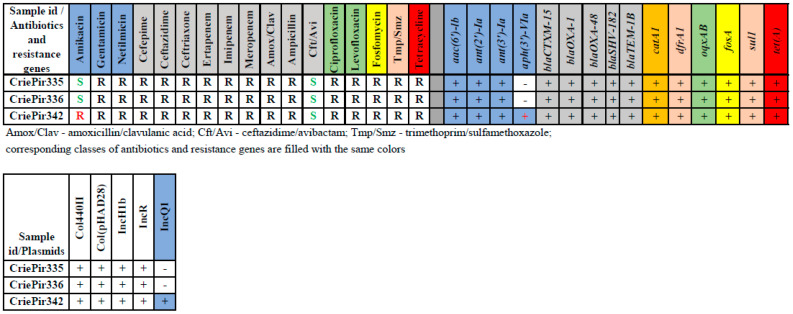
Phenotypic and genomic antibiotic resistance profiles of clinical *K. pneumoniae* isolates obtained from COVID-19 patients.

**Figure 2 antibiotics-11-01364-f002:**
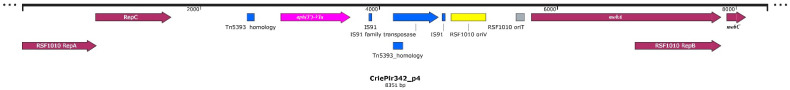
The structure of IncQ1 plasmid carrying amikacin resistance gene.

**Figure 3 antibiotics-11-01364-f003:**
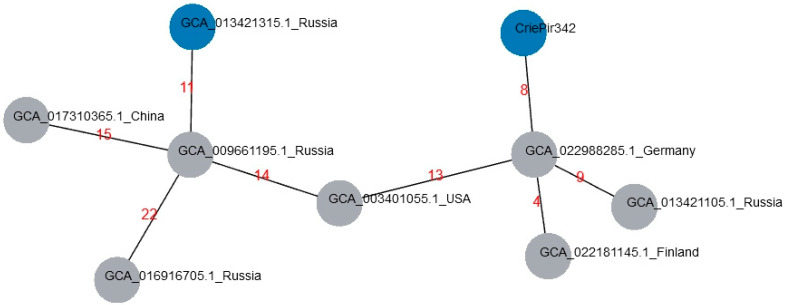
Minimum spanning tree based on cgMLST profiles for reference *K. pneumoniae* isolates and CriePir342. Since all our isolates possessed the close profiles, only CriePir342 is indicated. Red numbers show the number of allele differences between the corresponding isolates. The isolates carrying IncQ1 plasmid are colored in blue.

**Table 1 antibiotics-11-01364-t001:** Plasmid data for the clinical *K. pneumoniae* isolates studied.

Id	Length	RepliconType	RelaxaseFamily	Plasmid Type
2	282,773	IncH	MOB_H_, MPF_T_	Conjugative
3	74,680	IncR	-	Non-mobilizable
4	8351	IncQ	MOB_Q_	Mobilizable
5	5010	ColRNAI	-	Non-mobilizable
6	4052	ColRNAI	MOB_P_	Mobilizable
7	3511	ColRNAI	MOB_P_	Mobilizable

**Table 2 antibiotics-11-01364-t002:** Description of important virulence gene clusters in clinical *K. pneumoniae* isolates studied.

Gene Cluster	Function	Location
*acrAB*	efflux pump genes	Chromosome
*fimABCDEFGHIK*	fimbria production and biofilm formation	Chromosome
*entABCEF, fepABCDG*	enterobactin biosynthesis (siderophore)	Chromosome
*irp1,2*	iron acquisition system	Chromosome
*iucABCD, iutA*	aerobactin cluster-iron acquisition system	IncH1B plasmid
*manBC*	promoters of capsule synthesis genes	Chromosome
*mrkABCDFHIJ*	fimbria production and biofilm formation	Chromosome
*rcsAB*	exopolysaccharide biosynthesis	Chromosome
*rmpA2*	regulator of mucoid phenotype	IncH1B plasmid
*wbbMNO*	lipopolysaccharide synthesis	Chromosome
*ybtAEPQSTU*	yersiniabactin cluster-iron acquisition system	Chromosome

**Table 3 antibiotics-11-01364-t003:** Reference isolates from Genbank, with closest matches to the isolates studied according to cgMLST analysis.

Genbank Acc.	Number of Allele Differences	Country of Isolation	Isolate Collection Year
GCA_022988285.1	8	Germany	2016
GCA_022181145.1	10	Finland	2018
GCA_003401055.1	13	USA	2013
GCA_013421105.1	15	Russia	2018
GCA_009661195.1	17	Russia	2018
GCA_013421315.1	18	Russia	2018
GCA_017310365.1	18	China	2015

## Data Availability

Genome assemblies were submitted to NCBI Genbank under the project PRJNA839643. Accession numbers are as follows: JAMXKW000000000 (CriePir335), JAMXKV000000000 (CriePir336), JAMXKU000000000 (CriePir342).

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
