# Peer review of "Long-Read Whole Genome Sequencing Elucidates the Mechanisms of Amikacin Resistance in Multidrug-Resistant Klebsiella pneumoniae Isolates Obtained from COVID-19 Patients"

_antibiotics, 2022, doi:10.3390/antibiotics11101364_

Round 1
Reviewer 1 Report
This paper investigates the resistance profile and acquisition of Amikacin Resistance (aminoglycoside) in multidrug resistant Klebsiella pneumoniae isolates Obtained from COVID-19 Patients using WGS and phenotypic analysis. And the topic is original as it examines the possible further rise of K. pneumoniae carbapenem-resistance within the background of COVID-19 and possible acquisition mechanisms (mobilizable plasmid) through WGS. Even though the sample numbers are small it does further the field on AMR acquisition in K. pnuemonaie, the added novelty being COVID19 isolates. The methodology used was appropriate for this work – a combination of WGS and phenotypic analysis. Very interesting work and nicely written.
Some minor comments:
Figure 2 gene names need to be in italics, same with line 111, 115, 117.
Check that gene names are in italics throughout manuscript,Line 175 RmpA not rmpA.
Lines 223-225 need to be reworded as
The current study is limited to three isolates only since the main goal was to elucidate the difference in the phenotypic resistance profiles for the three K. pneumoniae isolates obtained in the same ward during this limited period. This was achieved with the help of third-generation sequencing.
Author Response
This paper investigates the resistance profile and acquisition of Amikacin Resistance (aminoglycoside) in multidrug resistant Klebsiella pneumoniae isolates Obtained from COVID-19 Patients using WGS and phenotypic analysis. And the topic is original as it examines the possible further rise of K. pneumoniae carbapenem-resistance within the background of COVID-19 and possible acquisition mechanisms (mobilizable plasmid) through WGS. Even though the sample numbers are small it does further the field on AMR acquisition in K. pnuemonaie, the added novelty being COVID19 isolates. The methodology used was appropriate for this work – a combination of WGS and phenotypic analysis. Very interesting work and nicely written.
Some minor comments:
Figure 2 gene names need to be in italics, same with line 111, 115, 117.
Check that gene names are in italics throughout manuscript,
The figure was fixed, and we checked the gene names throughout the manuscript
Line 175 RmpA not rmpA.
Fixed
Lines 223-225 need to be reworded as
The current study is limited to three isolates only since the main goal was to elucidate the difference in the phenotypic resistance profiles for the three K. pneumoniae isolates obtained in the same ward during this limited period. This was achieved with the help of third-generation sequencing.
Revised as suggested
Reviewer 2 Report
- Since the patients being Covid-19 does not have a special importance in the study, it does not need to be included in the title.
- Line 251-252: It cannot be said that MIC was detected by the Vitek method. Vitek 2 is an automated system that displays the estimated MIC range. It can be said that “ ….. by VITEK 2 automated system (bioMerieux, Marcy-252 l’Étoile, France)”.
Author Response
Since the patients being Covid-19 does not have a special importance in the study, it does not need to be included in the title.
Although the COVID-19 condition was not discussed in the manuscript as a main topic, we still think that mentioning it in the title is important since our work provides additional data regarding the change and acquisition of AMR by pathogenic bacteria in COVID ICU. Currently, there is still a global need in accumulation of such data, since the existing information is often controversial. We believe that adding COVID-19 to the title will help potential readers to find this information.
- Line 251-252: It cannot be said that MIC was detected by the Vitek method. Vitek 2 is an automated system that displays the estimated MIC range. It can be said that “ ….. by VITEK 2 automated system (bioMerieux, Marcy-252 l’Étoile, France)”.
Revised according to suggestions